# Selection of Salt-Tolerance and Ester-Producing Mutant *Saccharomyces cerevisiae* to Improve Flavour Formation of Soy Sauce during Co-Fermentation with *Torulopsis globosa*

**DOI:** 10.3390/foods12183449

**Published:** 2023-09-15

**Authors:** Kun-Qiang Hong, Xiao-Meng Fu, Fen-Fen Lei, Dong Chen, Dong-Ping He

**Affiliations:** 1College of Food Science and Engineering, Key Laboratory for Deep Processing of Major Grain and Oil, Ministry of Education, Hubei Key Laboratory for Processing and Transformation of Agricultural Products, Wuhan Polytechnic University, Wuhan 430023, China; 2Key Laboratory of Edible Oil Quality and Safety for State Market Regulation, Wuhan 430023, China; 3School of Chemical Engineering and Technology, Tianjin University, Tianjin 300350, China

**Keywords:** soy sauce fermentation, ARTP mutagenesis, co-fermentation, flavour, *Torulopsis*, *Saccharomyces cerevisiae*, *ATF1*

## Abstract

Soy sauce, as a traditional seasoning, is widely favoured by Chinese and other Asian people for its unique colour, smell, and taste. In this study, a salt-tolerance *Saccharomyces cerevisiae* strain HF-130 was obtained via three rounds of ARTP (Atmospheric and Room Temperature Plasma) mutagenesis and high-salt based screening. The ethanol production of mutant HF-130 was increased by 98.8% in very high gravity fermentation. Furthermore, *ATF1* gene was overexpressed in strain HF-130, generating ester-producing strain HF-130-ATF1. The ethyl acetate concentration of strain HF-130-ATF1 was increased by 130% compared to the strain HF-130. Finally, the soy sauce fermentation performance of *Torulopsis globosa* and HF-130-ATF1 was compared with *T. globosa*, HF-130, HF-130-ATF1, and Torulopsis and HF-130. Results showed ethyl acetate and isoamyl acetate concentrations in co-fermentation of *T. globosa* and HF-130-ATF1 were increased by 2.8-fold and 3.3-fold, respectively. In addition, the concentrations of ethyl propionate, ethyl caprylate, phenylethyl acetate, ethyl caprate, isobutyl acetate, isoamyl alcohol, phenylethyl alcohol, and phenylacetaldehyde were also improved. Notably, other three important flavour components, trimethylsilyl decyl ester, 2-methylbutanol, and octanoic acid were also detected in the co-fermentation of *T. globosa* and HF-130-ATF1, but not detected in the control strain *T. globosa*. This work is of great significance for improving the traditional soy sauce fermentation mode, and thus improving the flavour formation of soy sauce.

## 1. Introduction

Soy sauce, as a traditional fermented condiment, has a history of more than 2000 years in China [1,2]. Soy sauce exhibits unique colour, fragrance, and taste, which is obtained by using soybean and wheat as raw materials, using various enzymes secreted by microorganisms to hydrolyse plant proteins and carbohydrates [3,4]. Alcohols, aldehydes, esters, ketones, and others produced during fermentation give soy sauce a unique flavour [1,5]. Esters with relatively large total amount are one of the main aroma components of soy sauce [6]. Ethyl acetate is the main ester [7], which give soy sauce a strong flavour of ester aroma. Alcohols not only participate in the formation of the colour and flavour of [8], but also have certain bacteriostatic effectin soy sauce [9].

Currently, soy sauce fermentation technologies include low-salt solid fermentation and high-salt liquid fermentation. Low salt fermentation has been favoured by many enterprises for many years because of its advantages of small investment, short production cycle and simple technology [10]. However, it is difficult to make full use of the raw materials by low-salt solid fermentation, and the flavour compounds produced by this process is insufficient [11,12], which promote more and more large enterprises to choose high-salt liquid fermentation in China. High salt liquid fermentation can provide sufficient time for the slow formation of flavour substances in soy sauce. Moreover, the soy sauce produced by this process is of high quality and has become the mainstream in the brewing industry in Japan, South Korea and other places.

The formation of the unique flavour of soy sauce need the participation of various microorganisms [13,14,15], such as *Aspergillus oryzae*, yeast, lactobacillus, and other bacteria [16]. Yeast as the key host cell producing aroma, which directly affects the flavour production of soy sauce. In modern research, yeast with higher salt-tolerant was added in soy sauce fermentation to stimulate the accumulation of flavour substances and thus improving the quality of soy sauce. Many studies have shown that adding salt-tolerance *Candida versatilis* was a good choice to improve the flavour production in the process of soy sauce high-salt liquid fermentation [17]. The aroma components formed are mainly include HEMF (4-Hydroxy-2(or 5)-ethyl-5(or 2)-methyl-3(2H)-furanone) with sweet, caramel flavour and 4-EG (4-Ethylguaiacol) with smoky flavour [18,19]. When adding an extremely salt-tolerant *Hanses* strain in soy sauce fermentation, the ethyl acetate concentration was 734 times higher than that of *Torulopsis* yeast, at the same time, the production of major flavour components such as HEMF, 4-EG were also improved [20]. In addition, a *Candida versatilis* mutant S3-5 was isolated and added in fermentation, which improved the quality of soy sauce, shorten the fermentation cycle and reduced the fermentation costs [21].

Esters play an important role in flavouring, stabilizing, and seasoning, and the right dosage of esters contributes to enhancing the quality and taste of soy sauce [20,22]. The *ATF1* gene encoded alcohol acetyltransferase which affects a series of volatile esters formation. The expression level of *ATF1* gene is important for ethyl acetate and isoamyl acetate synthesis [23]. Zhang et al., used *S. cerevisiae* as the parent strain to construct a mutant with overexpressing the *ATF1* gene. The fermentation results showed that the ethyl acetate production increased by 22 times compared with the parent strain [24]. Therefore, ester-producing yeast strain with high salt-tolerance could be used to improve the flavour of soy sauce.

In this study, ARTP mutagenesis and high-salt based screening were used to obtain a salt-tolerance yeast strain. Further, the *ATF1* gene was overexpressed to improve ester production in the salt-tolerance yeast strain, generating strain HF-130-ATF1. Finally, the salt-tolerance and ester-producing strain HF-130-ATF1 was used in soy sauce fermentation with *T. globose* for enhancing flavour formation. The total concentration of alcohols, esters and aldehydes of soy sauce were significantly increased via co-fermenting HF-130-ATF1 and *T. globose*.

## 2. Materials and Methods

### 2.1. Strains, Plasmids, and Medium

Strains and plasmids were listed in Table 1. YPD (Yeast Extract Peptone Dextrose) medium consists of yeast extract 10 g/L, peptone 20 g/L, and glucose 20 g/L. The salt-added medium (YPDN) was based on the YPD medium, and different quality of NaCl was added according to the experimental requirements. Soy sauce fermentation media consists of cooked soybeans, peanut cake meal, and wheat bran (1:0.5:1, by weight), 120 g/L NaCl and tap water [25]. LB (Luria-Bertani, tryptone 10 g/L, yeast extract 5 g/L, and NaCl 10 g/L) medium with ampicillin (100 mg/L) was used for plasmid construction. SC-ura3 medium was prepared with glucose (20 g/L), yeast nitrogen base (6.7 g/L), and Ura-dropout powder (1.3 g/L), for selection of uracil prototrophic transformants. SC+5-FOA (5-Fluoroorotic acid) medium was used for selecting uracil auxotrophic transformants. The basic composition of the medium was the same as that of SC-ura3 medium, with the addition of 5-FOA solution (0.1 g/L). Agar powder (20 g/L) was added to all solid media used above. Primers were designed according to the gene sequence of *S. cerevisiae S288C* strain for the amplification and validation of fragments. The PCR primers were listed in Table 2. Yeast extract, peptone, and flavour substance standards were purchased from Sigma-Aldrich LLC. Shanghai, China. Yeast nitrogen base and 5-Fluoroorotic acid were purchased from Beijing Solarbio Science and Technology Co., Ltd. (Beijing, China). Other chemicals were purchased from Shanghai Shenggong Bioengineering Co., Ltd. (Shanghai, China).

### 2.2. ARTP Mutagenesis and High-Salt-Temperature Based Screening

Yeast cells (8–10 µL) were treated with pure helium plasma under 120 W radiofrequency power input and 10 SLM helium flow rate in ARTP Breeding Mutagenesis Machine (ARTP-M model, Yuanqing Tianmu Biotechnology Co., Ltd., Wuxi, China) [28]. Next, 1 mL of YPD medium was immediately added with treated cells. The resulting cell was cultured on YPDN medium at 30 °C for 48–72 h. All colony in the first round of screening was collected and treated with ARTP Mutagenesis Machine, and then mutants were cultured on YPDN plates at 30 °C for 48–72 h. Following the colonies in the second round of screening were collected and treated with ARTP Mutagenesis Machine, the mutants were cultivated on YPDN medium plates at 40 °C for 48–72 h for the third round of screening. The following Equation (1) was used to calculated cell lethality.
(1)$$Lethality(\%)=\frac{Control\;colonies-Survival\;colonies}{Control\;colonies}\times 100$$

The mutants in the third round of screening were chosen and inoculated into 50 mL YPD medium at 30 °C for 48 h, and then cell was washed to remove trace glucose. Following 0.50 g/L of dry cell mass were added into 100 mL of VHG medium (Very High Gravity medium, prepared by the method of Section 2.3 VHG fermentation) at 40 °C for 72 h [29].

### 2.3. VHG Fermentation

VHG fermentation medium was prepared by our previously reported method [29,30]. Notably, a final NaCl concentration of 120 g/L was added into VHG fermentation. Gas chromatography (GC) analysis was used to determine the abundance of esters and higher alcohol in VHG fermentation samples [26]. Detailed methods can also be found in the additional material.

### 2.4. Stress Tolerance Test

Dilution point plate test was used to determine the strain’s tolerance to different stress environments. Yeast strains were cultured to exponential phase in YPD medium. Before the dilution point plate test, strains OD_600_ was diluted to ~1. Then, strains were gradient diluted and coated on an acid- or salt-containing tolerance plate. When heat shock tolerance was measured, the experimental group was subjected to heat shock at 56 °C for 5 min. After a series of gradient dilutions, different amounts of yeast solution were spotted on the corresponding plates and cultured at 30 °C for 2–3 days. It should be noted that when comparing the thermal shock tolerance, both the heat-treated cells and the control cells were spotted on the YPD plate.

### 2.5. Growth Curves Test

The yeast cell growth curve under high salt concentration was measured in YPD liquid medium. A ring of yeast was selected from the slant medium and then cultured in YPD medium at 30 °C, 200 rpm for 12–16 h. Appropriate activated yeast solution was transferred to 50 mL YPDN medium and then cultured at 180 rpm, 40 °C or 180 rpm, 30 °C, respectively. Bioscreen Automated Growth Curves analysis system was used to plotted growth curves.

### 2.6. Construction of ATF1 Gene Overexpression Strain

Molecular techniques were used in nucleic acid manipulation [26]. Yeast transformation was carried out using the lithium acetate procedure as previously reported was carried out for replacing *URA3* gene (encoded orotidine-5’-phosphate decarboxylase) of strain HF-130 [31]. The fragment of *ura3* was amplified from AY12a-u strain genomic DNA using primer pairs URA3-F and URA3-R (Table 2). The fragment of *ura3* was introduced into the *URA3* gene site of HF-130, which screed on SC-ura plate to generate strain HF-130-u. For overexpressing *ATF1* gene, repairing dsDNA (donors, *HXT7p*, *ATF1*, and *HXT7t*) and pCas9-XII5.5 plasmid were prepared or constructed according to previously reported methodology [27], detailed steps can also be found in the additional material. Finally, the PCR products of *HXT7p-ATF1-HXT7t*, and plasmid pCas9-XII5.5 were added to yeast HF-130-u via electroporation method [27].

### 2.7. RT-qPCR Assay

*ATF1* transcriptional level assay was carried out according to our previously reported methodology [26]. Fungal mRNA out kit and Quant quantitative real-time PCR (qRT-PCR) (SYBR green) kit (Tiangen Biotech, Beijing, China) were used for mRNA isolation and mRNA relative quantification, respectively. Real-time PCR primer pairs were listed in Table 2. Threshold cycle (2^−ΔΔ*CT*^) method was used to calculate final data.

### 2.8. Soy Sauce Fermentation Experiment

Soy sauce fermentation was performed in cap-covered flasks with 100 mL medium at 35 °C. Yeast cell was cultured in YPD media at 30 °C until the OD_600_ was 2.0~2.5. Next, 10% seed medium was added into the above fermentation flasks. The flasks were shaken every 24 h. The soy sauce fermentation process was stopped when the residual sugar was not changed. HS-SPME-GC-MS (Headspace Solid Phase Microextraction Gas Chromatography-Mass Spectrometry) was used to extract flavour components from fermentation broth and analyze flavour components, detailed methods were described in the additional material [32]. The aroma compounds were determined by comparing the MS fragments detected with the NIST database (https://webbook.nist.gov/chemistry/, accessed on 15 March 2023). The quantitative analysis of the flavour compound was performed by calculation from the approximated curve using the linear least-squares method.

## 3. Results and Discussion

### 3.1. ARTP Mutagenesis and Mutant Screening

The lethality (%) of ARTP treatment to strain AY12a increased with the exposure time in a sigmoid pattern (Figure 1A). The lethality increased significantly from 25 s to 35 s and then the change was not noticeable. The lethality was 92.8% at 35 s of exposure time (Figure 1A), was chosen as the optimal exposure time in this work.

The growth of strain AY12a was inhibited when sodium chloride (NaCl) was added into the medium (Figure 1B). The surviving colony number was decreased significantly as sodium chloride concentration increased in each round screening (Figure 1B). In the first round of screening, the surviving colonies number of three plates supplemented with 60 g/L sodium chloride was 265, no colony was found in other plates supplemented with 90 g/L or 120 sodium chloride. All colony in the first round of screening was collected and used for the second round of ARTP mutagenesis. The surviving colony number was 1051 for 60 g/L sodium chloride, 324 in 90 g/L sodium chloride at 30 °C, and no colony number in 120 g/L sodium chloride. When the third round of ARTP mutagenesis was carried out, the surviving colony number in 120 g/L sodium chloride at 40 °C was 207, and its numbers in 60 g/L and 90 g/L sodium chloride at 40 °C were 2384 and 1185, respectively (Figure 1B).

A total of 207 mutants of third round of screening (named HF-1 to HF-207) and strain AY12a were subjected at 40 °C for ethanol fermentation. Ethanol production was detected to compare their fermentation ability. Among these, 73 mutants with higher ethanol production (Figure 1C), and the ethanol production of namely HF-130 was the highest of 130.26 g/L, 1.52-fold higher than that of control strain. The growth and ethanol production showed no significant difference after 25, 50, and 100 generations of HF-130 (Appendix A), indicated that mutant HF-130 was stable, therefore, used in further experiments.

### 3.2. Growth, Tolerance, and Fermentation Performance of HF-130

The growth performance of the mutant HF-130 and the parent strain AY12a was assayed at 30 °C in YPD medium or 40 °C in YPDN medium supplemented with 120 g/L sodium chloride, respectively. There were not any significant distinctions among the growth curves of all strains at 30 °C. When incubated at 40 °C in YPDN medium supplemented with 120 g/L sodium chloride, the OD_600_ maximum of HF-130 showed a 12.59% increase compared to the control strain AY12a, respectively (Figure 2A).

The salt content in high-salt liquid fermentation sauce mash is high, so the salt tolerance of added strain is very strict [33,34,35]. Yeast *Zygosaccharomyces rouxii* and *T. globosa* are commonly used in soy sauce fermentation [36,37,38,39]. Therefore, the tolerance performance of mutant HF-130 was test and compared with yeast *Zygosaccharomyces rouxii*, *T. globosa*, and AY12a. It is also important for yeast to tolerate other stress in soy sauce fermentation. Therefore, stress conditions including NaCl (120 g/L), heat shock (5 min at 56 °C), glucose (280 g/L glucose), and ethanol (80 g/L) of selected strains were tested on YPD plates. Results showed mutant HF-130 was markedly resistant to 56 °C thermal stress and 120 g/L of NaCl compared with AY12a (Figure 2B). Meanwhile, HF-130 also showed a stronger tolerance to 280 g/L of glucose and 80 g/L of ethanol than the parent strain. Compared with *Zygosaccharomyces rouxii* and *T. globosa*, mutant HF-130 also exhibited a stronger tolerance in these stress conditions. These results supported that HF-130 displayed the most desirable phenotype of elevated stress tolerance.

In addition, the fermentation performance, ethanol production, glucose consumption, organic acid and glycerol formation, loss of CO_2_, biomass and surviving rate were detected to assess HF-130 fermentation ability (Table 3). The free glucose concentration at the initiation of fermentation was ~279 g/L. The glucose consumption and ethanol production rates of HF-130 were no significant difference compared with AY12a in VHG fermentation at 30 °C, but that of HF-130 was improved in VHG fermentation supplemented with 120 g/L NaCl at 40 °C (Table 3). As shown in Table 3, the maximum amount of ethanol concentration was 116.70 g/L for AY12a, and 118.40 g/L for HF-130 30 °C. The residual glucose was 24.3 g/L for AY12a and 22.5 g/L for HF-130. HF-130 exhibited similar fermentation characteristics regarding CO_2_ loss, ethanol production, organic acid and glycerol formation, and residual sugar at 30 °C. Results showed that the fermentation performance of HF-130 was slightly differences compared with the parent strain. While fermented at 40 °C with 120 g/L NaCl, the glucose consumption rates for HF-130 was improved, and residual glucose concentration of HF-130 was 20.7 g/L, 2.1-fold lower than that of the control AY12a, revealing a high glucose utilization of HF-130 owning to the improved stress and heat tolerance. The biomass was 8.01 g/L for HF-130, 1.27-fold higher than that of AY12a. The surviving rate of HF-130 was 86.3%, 2.10-fold higher than 41.1% of AY12a. The final ethanol concentration of HF-130 was 130.26 g/L, 98.8% higher than 65.53 g/L of AY12a, exhibiting an increased ethanol production owning to the increase in biomass and surviving rate. The fermentation results revealed an elevation of HF-130 fermentation properties at 40 °C with high salt, suggesting that the mutant HF-130 was beneficial to increase the ethanol production in high stress condition.

### 3.3. Overexpressing of ATF1 Gene in HF-130

The above test results showed that HF-130 with high tolerance could be used in soy sauce fermentation. Though the content of flavour could be increased via adding yeast cell during soy sauce fermentation [40], this method required many optimization experiments. Previous research found alcohol acetyl transferase (AATase, encoded by *ATF1* gene) could catalyse the formation of acetate ester from alcohols and acetyl-CoA [41,42]. Therefore, *ATF1* gene was overexpressed in the mutant HF-130. Firstly, the *URA3* gene was replaced with *ura3* fragment, generating strain HF-130u, following *HXT7p-ATF1- HXT7t* expression modular was inserted into XII5.5 site via plasmid pCAS9-XII5.5, and then the gene expression of *URA3* was restored to construct the ester-producing strain HF-130-ATF1 (Figure 3A).

According to the test of growth curve and tolerance, there was no significant difference in the growth performance and stress tolerance between HF-130-ATF1 and HF-130 (Figure 3B,C). Therefore, the fermentation performance of HF-130-ATF1 was performed and compared with HF-130. The consumed glucose, the CO_2_ loss, organic acid production and alcohol production of HF-130-ATF1 were similar to the control strain HF-130. These results proved that overexpressing *ATF1* gene would not influence the fermentation properties of the resulting yeast HF-130-ATF1. However, ethyl acetate concentration of HF-130-ATF1 (58.32 mg/L) was increased by 1.3-fold compared to that of HF-130 (Figure 3D). These results were in agreement with the mRNA levels (Appendix A) and our previous research [26]. Isoamyl acetate was not detected in the fermentation sample. In addition, the concentrations of phenethyl alcohol, isobutyl alcohol, isoamyl alcohol and phenethyl alcohol were changed at different levels (Figure 3D). Fermentation results revealed that the overexpression of the *ATF1* gene in the HF-130 strain was beneficial for regulating the flavour production.

### 3.4. Co-Fermentation of Torulopsis and HF-130-ATF1 in Soy Sauce Fermentation

Soy sauce fermentation of the selected strain, HF-130-ATF1, HF-130, and *T. globosa* was carried out. The fermentation performance of *Torulopsis* and HF-130-ATF1 was compared with the *Torulopsis*, HF-130, HF-130-ATF1, and *Torulopsis* and HF-130. However, the fermentation activity and growth performance of the strain *Torulopsis* was very weak. Moreover, other two combinations of Torulopsis and HF-130-ATF1 and Torulopsis and HF-130 also grew slowly compared with the *S. cerevisiae* strain HF-130-ATF1 and HF-130 at 40 °C of the soy sauce fermentation temperature. This phenomenon might be due to the inhibition of the growth of *Torulopsis* in the high temperature (40 °C) and high stress (120 g/L NaCl) condition selected in this study. When 120 g/L of NaCl and lower temperature of 35 °C were chosen for soy sauce fermentation, the selected strains could grow in soy sauce fermentation. It was found that the residual glucose of *Torulopsis* and HF-130-ATF1 and *Torulopsis* and HF-130 was lower than that of the *Torulopsis*, HF-130, and HF-130-ATF1 at the end of fermentation. Residual glucose of *Torulopsis* and HF-130-ATF1 was 6.53 g/L, 3-fold lower compared with that of *Torulopsis* (Table 4).

The combined effect of flavour substances such as alcohols, esters, acids, phenols, ketones, and furanones forms soy sauce flavour. In this study, the main flavour components are presented in Table 4. The concentrations of ethyl acetate, isoamyl acetate, ethyl propionate, ethyl caprylate, phenylethyl acetate, ethyl caprate, isobutyl acetate, dimethyl ether, isoamyl alcohol, and 2-methylbutanol were distinctly improved when co-fermentation of *Torulopsis* and HF-130-ATF1 was carried out (Table 4). The total esters production was 131.76 mg/L, 2.02-fold higher than that of *Torulopsis* (Figure 4). The concentrations of the ethyl acetate (101.56 mg/L) and isoamyl acetate (4.78 mg/L) were increased by 2.8-fold and 3.3-fold compared with that of *Torulopsis*, respectively. Other esters were also detected, 1.02 mg/L for ethyl propionate, 2.69 mg/L for ethyl caprylate, 17.41 mg/L for phenylethyl acetate, 2.04 mg/L for ethyl caprate, and 0.59 mg/L for isobutyl acetate, which were 1.2-fold, 10.76-fold, 5.11-fold, 5.5-fold, and 1.5-fold higher than that of *Torulopsis.* In addition, the total higher alcohol concentration was increased by 32% (Figure 4). In addition, the isoamyl alcohol, phenylethyl alcohol, and phenylacetaldehyde increased to 2.52 mg/L, 6.07 mg/L, and 1.25 mg/L, respectively, which were 1.4-fold, 1.0-fold, and 6.6-fold higher, respectively, than that of *Torulopsis.* Notably, other three important flavour components, trimethylsilyl decyl ester (CAS No.: 55494-15-0) (1.58 mg/L), 2-methylbutanol (1.59 mg/L), and octanoic acid (1.82 mg/L), was detected in the fermentation of *Torulopsis* and HF-130-ATF1(Table 4). However, they were also found in other control fermentations containing *S. cerevisiae* cell, but not detected in the control strain *Torulopsis*.

Increased esters concentrations are the aim of improved soy sauce flavour. In this work, the enhanced ester production resulted in a decrease in ethanol production because ethanol along with some acids synthesized esters, which was the reason to the decrease in acetic acid. In addition, the decreased ethanol concentration would not affect soy sauce flavour because other alcohols also impart soy sauce alcohol. Fragrance. The increased phenylacetaldehyde can offset the decreased acetic acid, which can give soy sauce a floral aroma. As shown in Table 4, the flavour substances of co-fermentation *Torulopsis* and HF-130-ATF1 or single fermentation HF-130-ATF1 were generally higher than those of the corresponding of co-fermentation Torulopsis and HF-130-ATF1 or single HF-130. Furthermore, the co-fermentation Torulopsis and HF-130-ATF1 or Torulopsis and HF-130 contributed to improving the soy sauce flavour formation compared with the single fermentation *Torulopsis*. These results confirmed overexpression of *ATF1* gene in ARTP mutagenesis strain HF-130 and then co-cultured with *Torulopsis* can effectively improve the formation of soy sauce flavour substances, which is important for the food industry. However, more research needs be focused on the gene-edited-strain fermented foods and its safety for consumer acceptance.

## 4. Conclusions

ARTP mutagenesis was successfully improved the salt tolerance of *S. cerevisiae*. *Torulopsi* was fermented with the selected strain, HF-130-ATF1 with overexpressing *ATF1* gene, increased the production of flavour components of ethyl acetate, isoamyl acetate, ethyl caprylate, phenylethyl acetate, ethyl caprate, and phenylacetaldehyde, which formed characteristic natural aroma compounds instead. HF-130-ATF1 improved the main flavour components and enhanced the quality of soy sauce. Moreover, HF-130-ATF1 increased utilization of sugar and shortened the time, thus relatively decreasing the fermentation cost.

## Figures and Tables

**Figure 1 foods-12-03449-f001:**
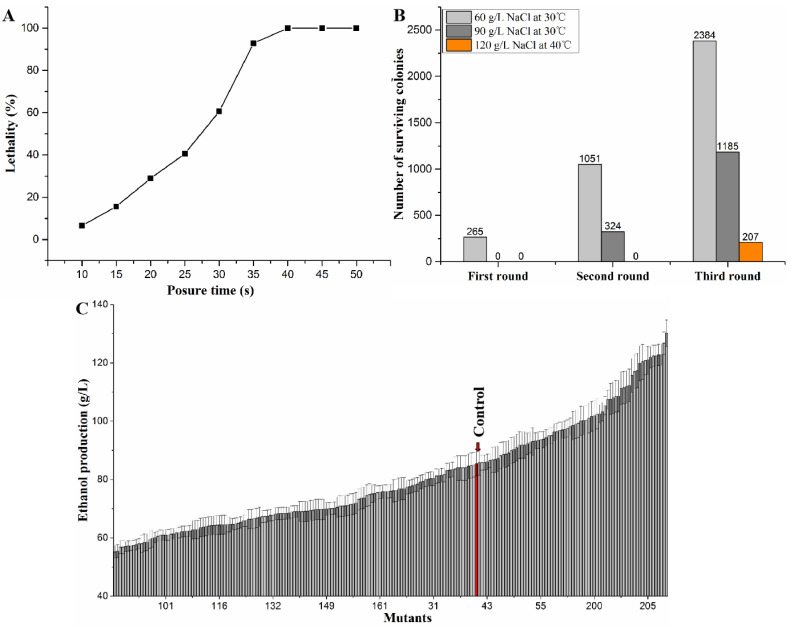
ARTP mutagenesis and screening of mutants. (**A**) Death rate curve of AY12a; (**B**) surviving colonies total number of three plates in each round of screening. The first two round of screening was carried out in YPDN medium supplemented (60, 90, and 120 g/L) sodium chloride at 30 °C. The third round of screening was carried out in YPDN medium supplemented (60, 90, and 120 g/L) sodium chloride at 40 °C. (**C**) Ethanol production of 207 mutants of third round of screening (named HF-1 to HF-207) and AY12a. Strains were fermented in VHG medium supplemented with 120 g/L NaCl at 40 °C. All the data are the average values of three independent experiments.

**Figure 2 foods-12-03449-f002:**
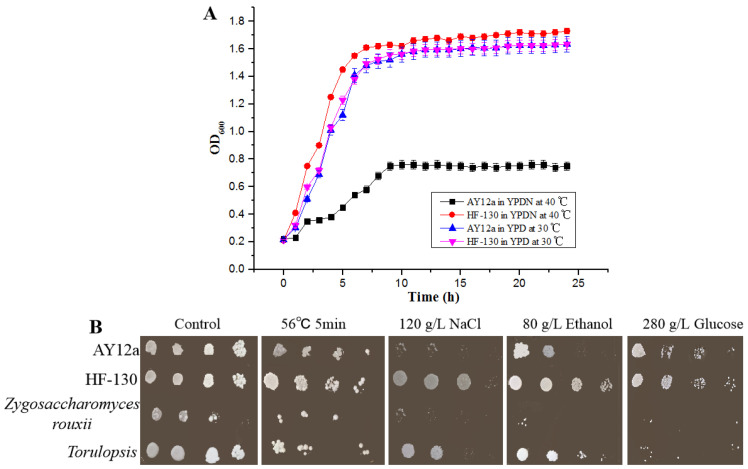
Growth and fermentation performance of HF-130. (**A**) Growth curves of mutant HF-130 and the parent strain AY12a. Mutant HF-130 and strain AY12a were assayed at 30 °C in YPD medium or 40 °C in YPDN liquid medium supplemented with 120 g/L sodium chloride, respectively. (**B**) Serial dilution assay of the selected mutant HF-130, *Zygosaccharomyces rouxii*, *T. globosa*, and AY12a, on the YPD plates (Control) and the YPD plates containing NaCl (120 g/L), heat shock (5 min at 56 °C), glucose (280 g/L glucose), and ethanol (80 g/L).

**Figure 3 foods-12-03449-f003:**
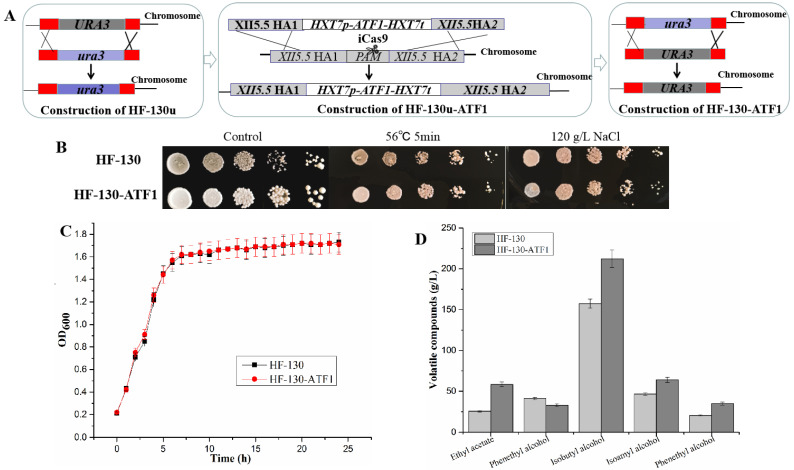
**Construction of HF-130-ATF and its growth, tolerance, and fermentation performance test.** (**A**) *ATF1* gene was overexpressed in mutant HF-130. the *URA3* gene was replaced with *ura3* fragment, generating strain HF-130u, following *HXT7p-ATF1-HXT7t* expression modular was inserted into XII5.5 site via plasmid pCAS9-XII5.5, generating strain HF-130u-ATF1, and then the gene expression of *URA3* was restored to construct the ester-producing strain HF-130-ATF1. (**B**) Serial dilution assay of mutant HF-130 and HF-130-ATF1, on the YPD plates (Control) and the YPDN plates containing NaCl (120 g/L), and heat shock (5 min at 56 °C). (**C**) Growth curves of mutant HF-130 and HF-130-ATF1. The two strains were tested at 40 °C in YPDN medium supplemented with 120 g/L sodium chloride, respectively. (**D**) The volatile compounds of HF-130 and HF-130-ATF1 in VHG medium.

**Figure 4 foods-12-03449-f004:**
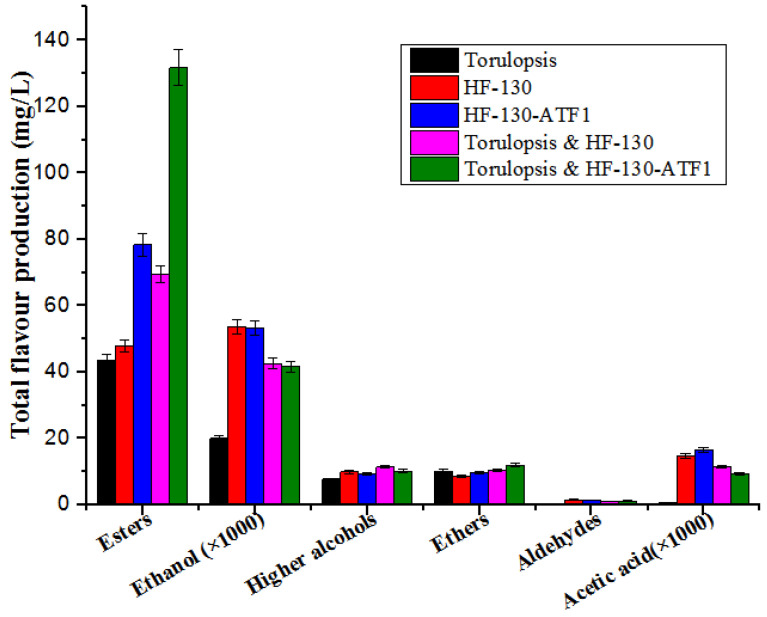
**Total flavour production of parental strains and engineered strain in soy sauce fermentation.** The total esters production includes the production of ethyl acetate, isoamyl acetate, ethyl propionate, ethyl caprylate, phenylethyl acetate, ethyl caprate, isobutyl acetate, and trimethylsilyl decyl ester. The total Higher alcohols production includes the production of isoamyl alcohol, phenylethyl alcohol, and 2-Methylbutanol.

**Table 1 foods-12-03449-t001:** Strains and plasmids used in this study.

Strain	Genotype	Source or Reference
AY12a	*MATa*	Angel Yeast Co., Ltd. (Wuhan, China)
HF-130	*MATa*, mutant obtained via ARTP Mutagenesis	This study
*Zygsoaccharomyces rouxii*	*MAT*	This lab
*Torulopsis globosa*	*MAT*	This lab
AY12a-u	*MATa ura3*	[26]
HF-130u	*MATa ura3*	This study
HF-130-ATF1	*MATa*, *XII5.5::PGK1*p-*ATF1*-*PGK1_t_*	This study
Plasmids		
pCas9	*2μ ampR TEF1p-iCas9 SNR52p*	[27]
pCas9-XII5.5	pCas9, containing 20-bp of XII5.5 gRNA sequence for gene insertion	This study

**Table 2 foods-12-03449-t002:** PCR primers used in this study.

Primers	Sequence (5′-3′)
XII5.5-F	CCGAAGTTGATTTGCTTCAAACC
XII5.5-R	CCCGAAATTGTTCCTACGAGAGCACATTTCGCCCGTTCC
HXT7P-F	GAGCGGGAACGGGCGAAATGTGCTCTCGTAGGAACAATTTCGGGCC
HXT7P-R	GATTTTTCTCATCGATTTCATTCATGGATCCACTAGTTCTAGATTTTTGATTAAAATTAAAAAAAC
ATF1-UF	GTTTTTTTAATTTTAATCAAAAATCTAGAACTAGTGGATCCATGAATGAAATCGATGAGAAAAATC
ATF1-DR	TATCGAATTCCTGCAGCCCGGGCTAAGGGCCTAAAAGGAGAGCTTTG
HXT7T-F	GCTCTCCTTTTAGGCCCTTAGCCCGGGCTGCAGGAATTCGATATC
HXT7T-R	GTAGCCAAGACCAAATAATACGTCGACGCAAGAACCATAATCCTC
XII5.5-U	GGAGGATTATGGTTCTTGCGTCGACGTATTATTTGGTCTTGGCTACTGGC
XII5.5-D	CTTTCTAGACTGATAGTTGGAGGCAC
URA3-F	GACCATCAAAGAAGGTTAATGTGGCTGTGGTTTCAGGG
URA3-R	CTTGATTTGTGCCCCGTAAAATACTGTTACTTGG
RTATF1-F	GGGTTCAATATACAAGGCTTCG
RTATF1-R	GCATCGGGCTCCTCTAACTG
ACT1-F	TTATTGATAACGGTTCTGGTATG
ACT1-R	CCTTGGTGTCTTGGTCTAC

**Table 3 foods-12-03449-t003:** Performance parameters of VHG fermentation ^a^.

Parameter	Fermentation at 30 °C without NaCl		Fermentation at 40 °C with 120 g/L NaCl	
	AY12-G	HF-130	AY12-G	HF-130
Ethanol production/(g/L)	116.70 ± 0.72	118.40 ± 0.91	65.53 ± 0.64	130.26 ± 0.79
Residual glucose/(g/L)	24.3 ± 1.0	22.5 ± 1.2	43.0 ± 0.8	20.7 ± 0.8
Glycerol/(g/L)	9.91 ± 0.36	9.94 ± 0.49	7.61 ± 0.37	10.13 ± 0.42
Organic acid ^b^/(g/L)	3.45 ± 0.25	3.56 ± 0.28	2.58 ± 0.20	3.72 ± 0.24
Loss of CO_2_/(g/L)	121.46 ± 0.68	122.79 ± 0.71	77.68 ± 0.59	128.53 ± 0.63
Biomass/(g/L)	7.89 ± 0.42	7.95 ± 0.40	5.28 ± 0.29	8.01± 0.32
Total glucose/(g/L)	279.42 ± 1.40	279.40 ± 1.56	279.11 ± 1.23	279.56 ± 1.47
Cell surviving rate in 48 h/(%)	Not test	Not test	41.1 ± 2.7	86.3 ± 1.2
Fermentation time/(h)	84	84	106	72
Productivity/(g/L/h)	1.389 ± 0.009	1.410 ± 0.011	0.679 ± 0.010	1.810 ± 0.009

^a^ Data are presented as the means and standard deviations of three independent experiments. ^b^ Specifically referring to acetic acid, succinic acid, and pyruvic acid.

**Table 4 foods-12-03449-t004:** Determination of parameters and flavour production in soy sauce fermentation with parental strains and engineered strain ^ab^.

Parameter ^d^	Torulopsis	HF-130 ^e^	HF-130-ATF1 ^e^	Torulopsis and HF-130 ^e^	Torulopsis and HF-130-ATF1 ^e^
Residual glucose/(g/L)	19.62 ± 0.84	10.36 ± 0.51	10.26 ± 0.43	6.64 ± 0.21 *	6.53 ± 0.32 *
Ethanol production/(g/L)	20.11 ± 1.12	53.56 ± 2.15 **	53.25 ± 1.58 **	42.65 ± 1.35 **	41.62 ± 2.01 **
Ethyl acetate/(mg/L)	36.93 ± 0.65	24.65 ± 0.23	53.26 ± 0.41 **	42.78 ± 0.71 **	101.56 ± 1.01 **
Isoamyl acetate/(mg/L)	1.45 ± 0.08	1.56 ± 0.10	2.15 ± 0.11	2.57 ± 0.12 *	4.87 ± 0.19 *
Ethyl propionate/(mg/L)	0.85 ± 0.01	0.64 ± 0.01	0.81 ± 0.01	0.84 ± 0.01	1.02 ± 0.01
Ethyl caprylate/(mg/L)	0.25 ± 0.02	2.12 ± 0.14 **	2.25 ± 0.16 **	2.32 ± 0.21 **	2.69 ± 0.22 **
Phenylethyl acetate/(mg/L)	3.41 ± 0.16	16.54 ± 0.56 **	17.15 ± 0.63 **	17.91 ± 0.73 **	17.41 ± 0.59 **
Ethyl caprate/(mg/L)	0.37 ± 0.05	1.25 ± 0.09 **	1.15 ± 0.08 **	1.76 ± 0.10 **	2.04 ± 0.15 **
Isobutyl acetate/(mg/L)	0.39 ± 0.05	0.33 ± 0.05	0.45 ± 0.01	0.34 ± 0.02	0.59 ± 0.05 *
Dimethyl ether/(mg/L)	10.23 ± 0.25	8.56 ± 0.63	9.66 ± 0.69	10.44 ± 0.11	11.96 ± 0.84
Trimethylsilyl decyl ester/(mg/L)	/^c^	0.80 ± 0.08	1.11 ± 0.12	1.07 ± 0.11	1.58 ± 0.09
Acetic acid/(g/L)	0.56 ± 0.09	14.74 ± 0.15 **	16.55 ± 0.21 **	11.56 ± 0.22 **	9.26 ± 0.16 **
Isoamyl alcohol/(mg/L)	1.84 ± 0.12	1.56 ± 0.12	2.61 ± 0.10	2.27 ± 0.15 *	2.52 ± 0.13 *
phenylethyl alcohol/(mg/L)	5.88 ± 0.51	6.89 ± 0.57	4.55 ± 0.85	7.6 ± 0.59 *	6.07 ± 0.85 *
Phenylacetaldehyde/(mg/L)	0.19 ± 0.08	1.56 ± 0.02 **	1.36 ± 0.08 **	1.05 ± 0.04 **	1.25 ± 0.01 **
2-Methylbutanol/(mg/L)	/^c^	1.45 ± 0.05	2.14 ± 0.09	1.53 ± 0.05	1.59 ± 0.07
1-Octadecene/(mg/L)	0.05 ± 0.01	0.06 ± 0.01	/^c^	0.05 ± 0.01	0.05 ± 0.01
Octanoic acid/(mg/L)	/^c^	2.01 ± 0.08	2.11 ± 0.05	1.92 ± 0.01	1.82 ± 0.04
Fermentation time/(day)	15	13	13	10	10

^a^ Temperature of Soy sauce fermentation was 35 °C; ^b^ Data are presented as the means and standard deviations of three independent experiments. ^c^ The flavour compound was not detected. ^d^ RI and odour description of flavour compounds were shown in Appendix A. ^e^ Student’s *t*-test *(* p* < 0.05, ** *p* < 0.01).

## Data Availability

The data used to support the findings of this study can be made available by the corresponding author upon request.

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
