# Peer review of "Selection of Salt-Tolerance and Ester-Producing Mutant *Saccharomyces cerevisiae* to Improve Flavour Formation of Soy Sauce during Co-Fermentation with *Torulopsis globosa"

_foods, 2023, doi:10.3390/foods12183449_

Round 1
Reviewer 1 Report
The manuscript "Co-fermentation of Torulopsis globosa and salt-tolerant, ester-producing Saccharomyces cerevisiae to improve the flavour formation of soy sauce" present novelty results with valuable application in the production of soy sauce. The work is very well writting but some revision are needed to improve de manuscript.
First, the title may be change to be more representative of se manuscript, por example “Selecction of salt-tolerance and ester-producing mutant Saccharomyces cerevisiae to improve flavor of soy sauce during co-fermentation with Torulopsis globosa”.
Line 61 y 64: define HEMF and $-EG
Line 76: In the final of introduction explain better de aim of the work and the conditions ssayed, avoiding mention of the conclusions.
Line 123: define VHG or add “see section 2.3”
Line 326: HF-130 increased salt or heat tolerance, check
Line 377 and 386 what is HF-130-9???
Line 410: deleted or detected?
Add information of trimethylsilyl decyl ester, is previously describe in other fermented food and beverage? Do you know what are the flavor notes of this compound??
Author Response
Dear editor and reviewers,
Thank you for your advice and comments concerning our manuscript entitled "Selection of salt-tolerance and ester-producing mutant Saccharomyces cerevisiae to improve flavor formation of soy sauce during co-fermentation with Torulopsis globosa"(foods-2609496). Those comments are valuable and very helpful for revising and improving our paper, as well as important guiding significance to our research. The manuscript has been revised and resubmitted with new lines and page numbers in the text. Revised parts are marked in red in the manuscript, and the comments are numbered for convenience. We appreciate editors/reviewers’ warm work earnestly, and hope that the correction will meet with approval. The main corrections in the paper and the responses to the reviewers’ or editor’s comments are as follows:
Reviewer #1:
The manuscript "Co-fermentation of Torulopsis globosa and salt-tolerant, ester-producing Saccharomyces cerevisiae to improve the flavour formation of soy sauce" present novelty results with valuable application in the production of soy sauce.The work is very well writing but some revision are needed to improve de manuscript.
Comment 1: First, the title may be change to be more representative of se manuscript, for example “Selecction of salt-tolerance and ester-producing mutant Saccharomyces cerevisiae to improve flavor of soy sauce during co-fermentation with Torulopsis globosa”.
Response 1: Thanks for your suggestion. We have revised the title to ‘Selection of salt-tolerance and ester-producing mutant Saccharomyces cerevisiae to improve flavor formation of soy sauce during co-fermentation with Torulopsis globosa’, which at lines 1-2 in revised manuscript.
Comment 2: Line 61 y 64: define HEMF and $-EG
Response 2: HEMF is 4-Hydroxy-2(or 5)-ethyl-5(or 2)-methyl-3(2H)-furanone, 4-EG is 4-ethylguaiacol. The information were added in revised manuscript, as can be seen at lines 64-65.
Comment 3: Line 76: In the final of introduction explain better de aim of the work and the conditions ssayed, avoiding mention of the conclusions.
Response 3: We re-wrote the final of introduction:
“In this study, ARTP mutagenesis and high-salt based screening were used to obtain a salt-tolerance yeast strain. Further, the ATF1 gene was overexpressed to improve ester production in the salt-tolerance yeast strain, generating strain HF-130-ATF1. Finally, the salt-tolerance and ester-producing strain HF-130-ATF1 was used in soy sauce fermentation with T. globose to improve flavor formation of soy sauce. The total concentration of alcohols, esters and aldehydes of soy sauce were significantly increased via co-fermenting HF-130-ATF1 and T. globose.” which also added at lines 81-87.
Comment 4: Line 123: define VHG or add “see section 2.3”
Response 4: VHG is the abbreviation of “Very High Gravity”. We revised the sentence to “Following 0.50 g/L of dry cell mass were added into 100 mL of VHG medium (Very High Gravity medium, prepared by the method of section 2.3 VHG fermentation) at 40 oC for 72 h”. The sentence was added at lines127-129.
Comment 5: Line 326: HF-130 increased salt or heat tolerance, check.
Response 5: In our work, for the first and second round of mutant screening, the surviving colonies were selected from plates supplemented with 60 g/L, 90 g/L or 120 sodium chloride at 30 °C. When the third round of ARTP mutagenesis was carried out, the surviving colony number in 120 g/L sodium chloride at 40°C was 207, and its numbers in 60 g/L and 90 g/L sodium chloride at 40°C were 2384 and 1185, respectively (Figure 1B). HF-130 is one of the 207 mutants selected from plates supplemented with 120 sodium chloride at 40 °C (these content can be seen at 237-251). In addition, serial dilution assay of the selected mutant HF-130, Zygosaccharomyces rouxii, T. globosa, and AY12a, on the YPD plates (Control) and the YPD plates containing NaCl (120 g/L), heat shock (5 min at 56 °C) also showed HF-130 exhibited increased stress and heat tolerance (these content can be seen at 279-291). Therefore, the residual glucose concentration of HF-130 was 20.7 g/L, 2.1-fold lower than that of the control AY12a, revealing a high glucose utilization of HF-130 owning to the increased stress and heat tolerance.
Comment 6: Line 377 and 386 what is HF-130-9???
Response 6: “HF-130-9” has been replaced by “HF-130-ATF1”, at lines 364 and 373.
Comment 7: Line 410: deleted or detected?
Response 7: “deleted” was repalced by “detected”, which was shown at line 400 in revised manuscript.
Comment 8: Add information of trimethylsilyl decyl ester, is previously describe in other fermented food and beverage? Do you know what are the flavor notes of this compound??
Response 8: Chemical Abstracts Service Registry Number (CAS No.) of trimethylsilyl decyl ester is 55494-15-0, the information was also added at lines 407-408. In addition, trimethylsilyl decyl ester exhibits odorless or slight fruity aroma and is currently mainly based on chemical synthesis (Rotzoll, Sven; Ullah, Ehsan; Goerls, Helmar; Fischer, Christine; Langer, Peter Tetrahedron, 2007, 63(12), 2647-2656; Kotowska, Urszula; Isidorov, Valery A. Central European Journal of Chemistry, 2011,9(5),813-824). The presence of trimethylsilyl decyl ester may be the result of a combination of chemical reactions of substrates in the pretreatment process or biochemical reactions during fermentation in this study.
We appreciate for Editors/Reviewers’ warm work earnestly.
Once again, thank you very much for your comments and suggestions.
Sincerely yours,
Hong Kun-qaing
Wuhan Polytechnic University, 68 Xuefu South Road, Changqing Garden, Wuhan, 430023, P. R. CHINA
Reviewer 2 Report
Line 14-28 Abstract: Both the research aim and the methodology to achieve it are not clear.
Abbreviation. Please write out the entire phrase first, then abbreviate it.
Line 68: Why are esters required to create a pleasant soy-sauce flavour?
Line 230: GC-MS analysis?
Line 234-235: Did you use SPME or direct solvent extraction to extract the samples?
Line 231-240: The SPME-GCMS method lack off
(a) a identification method: comparing their mass spectra with those contained in a mass spectral library, and/ or their Linear Retention Indices (LRI) with previously published literature (or with those authentic compounds),
(b) a SPME fiber spesification,
(c) a method of quantification. Did you run an internal standard or authentic compound into the GCMS for its quantification?
Line 233-234: This sentence is not clear. How much samples were added into the vial?
Line 372: The GC-MS analysis … ?
Line 395-396 Table 4: In the text or the supplementary section, a table containing a list of volatile compounds along with their LRI experiment, LRI literature, concentration, and odour description should be provided.
Line 404-407 and 433-435: Both sentences are not clear. What kinds of odour that esters impart to improve the soy-sauce flavour? As there is no sensory data, odour description of each (or esters) volatile compounds from literature (in a table) may support the argument.
Minor editing of English language required
Author Response
Dear editor and reviewers,
Thank you for your advice and comments concerning our manuscript entitled "Selection of salt-tolerance and ester-producing mutant Saccharomyces cerevisiae to improve flavor formation of soy sauce during co-fermentation with Torulopsis globosa"(foods-2609496). Those comments are valuable and very helpful for revising and improving our paper, as well as important guiding significance to our research. The manuscript has been revised and resubmitted with new lines and page numbers in the text. Revised parts are marked in red in the manuscript, and the comments are numbered for convenience. We appreciate editors/reviewers’ warm work earnestly, and hope that the correction will meet with approval. The main corrections in the paper and the responses to the reviewers’ or editor’s comments are as follows:
Reviewer #2:
Comment 1: Line 14-28 Abstract: Both the research aim and the methodology to achieve it are not clear.
Response 1: Thanks for your suggestion. Abstract section was revised as follows:
“In this study, a salt-tolerance Saccharomyces cerevisiae strain HF-130 was obtained via three rounds of ARTP mutagenesis and high-salt based screening. The ethanol production of mutant HF-130 was 98.8% higher than that of the control strain in very high gravity fermentation. Furthermore, ATF1 gene was overexpressed in strain HF-130, generating ester-producing strain HF-130-ATF1. The ethyl acetate concentration of strain HF-130-ATF1 was increased by 130 % compared to the strain HF-130. Finally, the soy sauce fermentation performance of Torulopsis globosa & HF-130-ATF1 was compared with T. globosa, HF-130, HF-130-ATF1, and Torulopsis & HF-130. Results showed the concentrations of the ethyl acetate and isoamyl acetate in co-fermentation of T. globosa and HF-130-ATF1 were increased by 2.8-fold and 3.3-fold, respectively. In addition, the concentrations of ethyl propionate, ethyl caprylate, phenylethyl acetate, ethyl caprate, isobutyl acetate, isoamyl alcohol, phenylethyl alcohol, and phenylacetaldehyde were also improved.” at lines 17-30.
Comment 2: Abbreviation. Please write out the entire phrase first, then abbreviate it.
Response 2: We have checked the abbreviations in the full text, and listed the entire phrase first, such as ARTP (Atmospheric and Room Temperature Plasma) at line 19, HEMF (4-Hydroxy-2(or 5)-ethyl-5(or 2)-methyl-3(2H)-furanone) and 4-EG (4-Ethylguaiacol) at lines 64-65, YPD (Yeast Extract Peptone Dextrose) at line 91, LB (Luria-Bertani)at line 96.
Comment 3: Line 68: Why are esters required to create a pleasant soy-sauce flavour?
Response 3: In general, the importance of esters to soy sauce is mainly reflected in the following aspects:
Aroma and flavor enhancement: esters often have aroma and flavor characteristics, which can give soy sauce a unique flavor and taste. For example, ethyl acetate is a commonly used aroma ester, and adding an appropriate amount to soy sauce can increase its aroma and flavor.
Soy sauce stability: esters can improve the stability of soy sauce and prevent its qualitative change. The esters in soy sauce can form a colloidal structure, increase the viscosity and consistency of soy sauce, and keep it for a long shelf life.
Seasoning: esters can play a seasoning role and improve the taste of soy sauce. Different esters have different active groups that can interact with other components in soy sauce to further enhance the flavor of soy sauce.
In this manuscript, the sentence “A high concentration of esters is necessary to develop good soy-sauce flavor” was replaced by “Esters play an important role in flavoring, stabilizing, and seasoning, and the right dosage of esters contributes to improve the quality and taste of soy sauce” at lines 72-73.
Comment 4: Line 230: GC-MS analysis? Line 234-235: Did you use SPME or direct solvent extraction to extract the samples? Line 231-240: The SPME-GCMS method lack off. (a) a identification method: comparing their mass spectra with those contained in a mass spectral library, and/ or their Linear Retention Indices (LRI) with previously published literature (or with those authentic compounds), (b) a SPME fiber spesification,(c) a method of quantification. Did you run an internal standard or authentic compound into the GCMS for its quantification? Line 233-234: This sentence is not clear. How much samples were added into the vial?
Response 4: Thanks for your questions. Those comments are interconnected. Here we responded both of these questions together. In this work, HS-SPME-GC-MS (Headspace Solid Phase Microextraction Gas Chromatography-Mass Spectrometry) was used to extract flavor components from soy sauce fermentation broth and analyze flavor components. Based on the comments of the reviewers, we described SPME-GCMS method in detail as follows:
“HS-SPME-GC-MS (Headspace Solid Phase Microextraction Gas Chromatography-Mass Spectrometry) was used to extract flavor components from fermentation broth and analyze flavor components[32].
HS-SPME Sampling Conditions 8 mL of sample was put into a 20 mL vial and spiked with 3 g NaCl, and a small magnetic stirrer was added. The sample was equilibrated for 10 min and extracted for 50 min at 60 °C with continuous stirring. The SPME fiber holder equipped with DVB/CAR/PDMS fiber (Supelco, Inc., Bellefonte, PA, USA) was used for aroma compounds extraction in this study. After extraction, the fiber was inserted into the injection port of a GC-MS system (at 250 °C for 5 min).
Identification and quantitative analysis Identification of flavour compound was carried out using an Agilent 7890A GC coupled with an Agilent 5975C mass selective detector (MSD). The sample was analyzed on a CP-Wax column (50 m x 250 μm inner diameter, 0.2 μm film thickness). The injector temperature was 250 °C and the split mode was used (ratio 15:1). The oven temperature was held at 50 °C for 3 min, raised to 70 °C at a rate of 3 °C/ min increased to 170 °C at a rate of 3 °C/min, then increased to 240 °C at a rate of 8 °C/min and held at 240 °C for 3 min. The column carrier gas was helium at a constant flow rate of 1 mL/min. The mass spectrometer was operated in electron-impact mode at 70 eV. The temperatures of the interface, ion source and quadrupole were 280, 230 and 150 °C, respectively. The aroma compounds were determined by comparing the MS fragments detected with the mass spectra present in the NIST MS spectral database (https://webbook.nist.gov/chemistry/). The compounds identified by MS were further confirmed by comparing the retention times generated for each reference compound analyzed, using a commercial hydrocarbon mixture (C8–C40) for determination of the retention indices (RI). The quantitative analysis of the flavour compound was performed by calculation from the approximated curve using the linear least-squares method. The respective quantitative values of the flavour compounds were determined by averaging the triplicate experiments.” at lines 200-225.
Comment 5: Line 372: The GC-MS analysis … ?
Response 5: Gas chromatography (GC) analysis was used to determine the abundance of esters and higher alcohol in VHG fermentation samples, and GC-MS analysis was used to determine the flavour in soy sauce fermentation. The relevant description was shown at lines 152-154, and lines 200-203.
Comment 6: Line 395-396 Table 4: In the text or the supplementary section, a table containing a list of volatile compounds along with their LRI experiment, LRI literature, concentration, and odour description should be provided.
Response 6: Thanks for your questions. RI experiment, RI literature and odour description of flavour compounds were shown in Table S1. The volatile compounds concentration were listed in the text. In addition, we have also marked it in the comments of Table 4, which can be seen at lines 383-390. RI and odour description of flavour compounds were shown in supplementary section Table S1.
Table S1 RI and Odour description of flavour compounds
|
Flavour compounds |
RI-EXa |
RI-LIb |
Odour descriptionc |
|
Ethanol |
454 |
448 |
alcoholic |
|
Ethyl acetate |
598.8 |
605 |
fruity flavor |
|
Isoamyl acetate |
862 |
866 |
fruity flavor |
|
Ethyl propionate |
998.4 |
977 |
fruity flavor |
|
Ethyl caprylate |
1191.9 |
1193.8 |
fruity flavor |
|
Phenylethyl acetate |
1254.8 |
1270 |
aromatic taste |
|
Ethyl caprate |
1369 |
1400 |
waxy |
|
Isobutyl acetate |
1001 |
1013 |
fruity flavor |
|
Dimethyl ether |
481 |
478 |
ether taste, sweet taste |
|
Trimethylsilyl decyl ester |
1446 |
1450 |
odorless,or light scent in low concentration |
|
Acetic acid |
623 |
645 |
sour |
|
Isoamyl alcohol |
1151 |
1195 |
mild odor; alcoholic |
|
phenylethyl alcohol |
1095 |
1100 |
floral odor of roses |
|
Phenylacetaldehyde |
1045 |
1043 |
aromatic taste |
|
2-Methylbutanol |
698 |
700 |
ethereal |
|
1-Octadecene |
1793 |
1799 |
mild hydrocarbon odor |
|
Octanoic acid |
2072 |
2070 |
pungent odor, fruity aroma in low concentration |
a Experimental value;
b Literature reported values in the NISTMS spectral database (https://webbook.nist.gov/chemistry/);
c Odour description refers to ChemicalBook (https://www.chemicalbook.com/)
Comment 7: Line 404-407 and 433-435: Both sentences are not clear. What kinds of odour that esters impart to improve the soy-sauce flavour? As there is no sensory data, odour description of each (or esters) volatile compounds from literature (in a table) may support the argument.
Response 7: Thanks for your questions. Odour description of each (or esters) volatile compounds were shown in supplementary section Table S1.
The sentence “It can be found that the flavour was distinctly improved by ethyl acetate, isoamyl acetate, ethyl propionate, ethyl caprylate, phenylethyl acetate, ethyl caprate, isobutyl acetate, dimethyl ether, isoamyl alcohol and 2-Methylbutanol when co-fermentation of Torulopsis and HF-130-ATF1 was carried out” was replaced by “It can be found that the concentration of ethyl acetate, isoamyl acetate, ethyl propionate, ethyl caprylate, phenylethyl acetate, ethyl caprate, isobutyl acetate, dimethyl ether, isoamyl alcohol and 2-methylbutanol was distinctly improved when co-fermentation of Torulopsis and HF-130-ATF1 was carried out” at lines 393-397.
The sentence “Furthermore, the co-fermentation Torulopsis & HF-130-ATF1 or Torulopsis & HF-130 contributed to improve the soy-sauce flavour compared with the single fermentation Torulopsis. These results confirmed overexpression of ATF1 gene in ARTP mutagenesis strain HF-130 and then co-cultured with Torulopsis was very helpful in improving the soy sauce flavor.” was replaced by “Furthermore, the co-fermentation Torulopsis & HF-130-ATF1 or Torulopsis & HF-130 contributed to improve the soy-sauce flavour formation compared with the single fermentation Torulopsis. These results confirmed overexpression of ATF1 gene in ARTP mutagenesis strain HF-130 and then co-cultured with Torulopsis can effectively improve the formation of soy sauce flavor substances, which is of great significance for the development of the food industry. However, more researches need be focused on the gene-edited-strain fermented foods and its safety for consumer acceptance. ” at lines 421-427.
In addition, we have also revised other sections, and the changes have also been red.
We tried our best to improve the manuscript and made some changes in the manuscript. These changes will not influence the content and framework of the paper.
We appreciate for Editors/Reviewers’ warm work earnestly.
Once again, thank you very much for your comments and suggestions.
Sincerely yours,
Hong Kun-qaing
Wuhan Polytechnic University, 68 Xuefu South Road, Changqing Garden, Wuhan, 430023, P. R. CHINA
Reviewer 3 Report
The effects of yeast modification, efficiency and chemical evaluation (alcohol, esters) of the resulting soy sauce are very interesting.
However, the work uses genetic mutations of microorganisms used in food production. Such food is often met with a lack of acceptance from consumers. This problem should be included in the literature part and discussion.
Others:
Line 81: what is “salt-tolerant ester-producing strain HF-130-9”? this is not clear at all. Is this the same as HF-130-ATF1? Please explain
All media, reagents need to be described (company/producer, city, country)
Line 228 : soy-sauce media, what is the composition of the media?
Table 4 - describe statistically - statistically significant differences
Line 428- where is Table 5? Did you mean Table 4?
English ok
Author Response
Dear editor and reviewers,
Thank you for your advice and comments concerning our manuscript entitled "Selection of salt-tolerance and ester-producing mutant Saccharomyces cerevisiae to improve flavor formation of soy sauce during co-fermentation with Torulopsis globosa"(foods-2609496). Those comments are valuable and very helpful for revising and improving our paper, as well as important guiding significance to our research. The manuscript has been revised and resubmitted with new lines and page numbers in the text. Revised parts are marked in red in the manuscript, and the comments are numbered for convenience. We appreciate editors/reviewers’ warm work earnestly, and hope that the correction will meet with approval. The main corrections in the paper and the responses to the reviewers’ or editor’s comments are as follows:
Reviewer #3:
The effects of yeast modification, efficiency and chemical evaluation (alcohol, esters) of the resulting soy sauce are very interesting.
Comment 1: The work uses genetic mutations of microorganisms used in food production. Such food is often met with a lack of acceptance from consumers. This problem should be included in the literature part and discussion.
Response 1: Thanks for your questions.
With the development of biotechnology, gene editing tools in the field of food research can effectively help people achieve the expected goals, such as the flavor substance regulation of beer, liquor and other fermented food. In addition, gene editing tools was also successfully applied in some oral drugs and its precursors biosynthesis, as well as the United States genetically modified soybeans and corn research. In this study, the overexpression of ATF1 in the screened mutant strain can effectively improve the synthesis of soy sauce flavor substances, which is of great significance for the development of the food industry. However, researchers should admit that of more researches need be focused on the gene-edited-strain fermented foods and their safety for consumer acceptance.
Sentence “These results confirmed overexpression of ATF1 gene in ARTP mutagenesis strain HF-130 and then co-cultured with Torulopsis can effectively improve the formation of soy sauce flavor substances, which is of great significance for the development of the food industry. However, more researches need be focused on the gene-edited-strain fermented foods and its safety for consumer acceptance.” was added at lines 423-427.
Comment 2: Line 81: what is “salt-tolerant ester-producing strain HF-130-9”? this is not clear at all. Is this the same as HF-130-ATF1? Please explain
Response 2: “HF-130-9” has been replaced by “HF-130-ATF1”, at lines 364 and 373.
Comment 3: All media, reagents need to be described (company/producer, city, country)
Response 3: All media, reagents were described at lines 105-108.
“Yeast extract, peptone and flavor substance standards were purchased from Sigma-Aldrich LLC. Shanghai, China. Yeast nitrogen base and 5-Fluoroorotic acid were purchased from Beijing Solarbio Science & Technology Co.,Ltd. Other chemicals were purchased from Shanghai Shenggong Bioengineering Co., Ltd.”
Comment 4:Line 228 :soy-sauce media, what is the composition of the media?
Response 4: Soy sauce fermentation media includes cooked soybeans, peanut cake meal and wheat bran (1:0.5:1, by weight), 120 g/L NaCl and tap water.
“Soy sauce fermentation media prepared with cooked soybeans, peanut cake meal and wheat bran (1:0.5:1, by weight), were supplemented with 120 g/L NaCl and tap water” has been described at lines 93-95.
Comment 5:Table 4 - describe statistically significant differences
Response 5: Statistically significant differences were described in Table4.
Comment 6:Line 428- where is Table 5? Did you mean Table 4?
Response 6: we have checked and revised it. “Table 5” was replaced by “Table 4” at line 418.
In addition, we have also revised other sections, and the changes have also been red.
We tried our best to improve the manuscript and made some changes in the manuscript. These changes will not influence the content and framework of the paper.
We appreciate for Editors/Reviewers’ warm work earnestly.
Once again, thank you very much for your comments and suggestions.
Sincerely yours,
Hong Kun-qaing
Wuhan Polytechnic University, 68 Xuefu South Road, Changqing Garden, Wuhan, 430023, P. R. CHINA
Reviewer 4 Report
The authors did a good job describing the investigation of co-fermentation of Torulopsis globosa and HF-130-ATF1. The work is well described and the results are supported by provided data. With a few exceptions such as line 377-378 which needs to be reworded, the paper was well written and the English was good.
Overall, I only saw a few issues with awkward wording which did not really take away from the quality of the manuscript. The one exception to this was in line 377-378 which needs to be reworded.
Author Response
Dear editor and reviewers,
Thank you for your advice and comments concerning our manuscript entitled "Selection of salt-tolerance and ester-producing mutant Saccharomyces cerevisiae to improve flavor formation of soy sauce during co-fermentation with Torulopsis globosa"(foods-2609496). Those comments are valuable and very helpful for revising and improving our paper, as well as important guiding significance to our research. The manuscript has been revised and resubmitted with new lines and page numbers in the text. Revised parts are marked in red in the manuscript, and the comments are numbered for convenience. We appreciate editors/reviewers’ warm work earnestly, and hope that the correction will meet with approval. The main corrections in the paper and the responses to the reviewers’ or editor’s comments are as follows:
Reviewer #4:
The authors did a good job describing the investigation of co-fermentation of Torulopsis globosa and HF-130-ATF1. The work is well described and the results are supported by provided data. With a few exceptions such as line 377-378 which needs to be reworded, the paper was well written and the English was good.
Response: Thank you reviewers for your approval. We have revised these sentences.
“However, the fermentation activity of the strain Torulopsis was very weak and the strain almost not grew, and other two combinations of Torulopsis & HF-130-ATF1 and Torulopsis & HF-130 also grew slowly compared with the S. cerevisiae strain HF-130-ATF1 and HF-130 at 40 °C of the soy sauce fermentation temperature.” was replaced by “However, the fermentation activity and growth performance of the strain Torulopsis was very weak. Besides, other two combinations of Torulopsis & HF-130-ATF1 and Torulopsis & HF-130 also grew slowly compared with the S. cerevisiae strain HF-130-ATF1 and HF-130 at 40 °C of the soy sauce fermentation temperature.” at lines 364-368.
In addition, we have also revised other sections, and the changes have also been red.
We tried our best to improve the manuscript and made some changes in the manuscript. These changes will not influence the content and framework of the paper.
We appreciate for Editors/Reviewers’ warm work earnestly.
Once again, thank you very much for your comments and suggestions.
Sincerely yours,
Hong Kun-qaing
Wuhan Polytechnic University, 68 Xuefu South Road, Changqing Garden, Wuhan, 430023, P. R. CHINA
Round 2
Reviewer 2 Report
The authors of the manuscript under the title ''Co-fermentation of Torulopsis globosa and salt-tolerant, ester-producing Saccharomyces cerevisiae to improve the flavour formation of soy sauce '' have followed all of suggestions.
The authors have described the method of GC-MS analysis properly and back up Table 4 (Determination of parameters and flavour production in soy sauce fermentation with parental strains and engineered strain) with its RI and odour description (Table S1).
Minor editing of English language is required.